# Earthquake Event Recognition on Smartphones Based on Neural Network Models

**DOI:** 10.3390/s22228769

**Published:** 2022-11-13

**Authors:** Meirong Chen, Chaoyong Peng, Zhenpeng Cheng

**Affiliations:** 1Institute of Geophysics, China Earthquake Administration, Beijing 100081, China; 2Key Laboratory of Earthquake Source Physics, China Earthquake Administration, Beijing 100081, China

**Keywords:** earthquake early warning, smartphone, seismic event recognition, neural network

## Abstract

Using sensors embedded in smartphones to study earthquake early warning (EEW) technology can effectively reduce the high construction and maintenance costs of traditional EEW systems. However, due to the impact of human activities, it is very difficult to accurately detect seismic events recorded on mobile phones. In this paper, to improve the detection accuracy of earthquakes on mobile phones, we investigated the suitability of different types of neural network models in seismic event detection. Firstly, we collected three-component acceleration records corresponding to human activities in various scenarios such as walking, running, and cycling through our self-developed mobile application. Combined with traditional strong-motion seismic event records fusing typical mobile phone accelerometer self-noise, all records were used for establishing the training and testing dataset. Finally, two types of neural network models, fully connected and convolutional neural networks, were trained, validated, and tested. The results showed that the accuracy rates of the neural network models were all over 98%, and the precision rate for seismic events and the recall rate for non-earthquake events could both reach 99%, indicating that the introduction of neural networks into the earthquake recognition on smartphones can significantly enhance the accuracy of seismic event recognition. Therefore, we can exceedingly reduce the amount of data transmitted to the processing server, further lowering the load on the server processor and effectively increasing the lead time at each target site for an EEW system.

## 1. Introduction

Earthquake early warning (EEW) is an effective means of earthquake disaster prevention commonly used in the world [1,2,3,4,5,6]. The existing domestic EEW method is mainly to deploy a complete and precise EEW system characterized by high integration and real-time response in a specific area, and uses the high-precision accelerometer built in the station to identify seismic events and determine whether an earthquake occurs or not. At the same time, due to the fixed location of the seismic station, the collected data are not affected by human events, so the seismic signal can be rapidly and accurately captured. However, deploying a traditional EEW system needs a huge investment; some countries and regions have then begun to use low-cost sensor devices for EEW and seismic intensity rapid reporting [7,8,9,10,11,12].

In recent years, with the popularization of smartphones and the development of Micro-Electro-Mechanical System (MEMS) technology, seismologists have found that it is feasible to take advantage of mobile phones to detect seismic events and conduct relevant analysis [13], and have made great progress in the seismology field [14,15,16,17,18,19,20,21,22]. In China, the mobile phone application (APP) is generally used as an early warning information release terminal, and it is unnecessary to gather acceleration and other data, so there is no need to take seismic event detection into consideration on mobile phones. Two examples are the Emergency Earthquake Information APP, officially developed by the Sichuan Earthquake Administration (http://www.scdzj.gov.cn/, accessed on 8 October 2022) and the ICL EEW APP, developed by the Chengdu High-tech Disaster Reduction Research Institute (http://www.365icl.com/index.asp, accessed on 8 October 2022). However, in few countries and regions, they not only use mobile phones as EEW information-issuing terminals, but also consider them as sensors for real-time data acquisition. If an earthquake is detected, the data will be immediately sent to the server for EEW processing. Typical APPs include MyShake, jointly developed by UC Berkeley and Deutsche Telekom [23,24,25], and Earthquake Network APP, developed by Finazzi [12,26]. The advantages of the latter mode are as follows:Making use of the broad distribution and vast amount of smartphones, we can increase the seismic station density, expand the early warning coverage, and reduce the investment in deploying and operating a traditional EEW system;With the Global Navigation Satellite System (GNSS) positioning function embedded in smartphones, the earthquake epicenter can be promptly located;APPs are more convenient to develop, update and maintain, and some EEW parameters can be customized according to user needs.

The most significant step in using mobile phones to carry out EEW is how to quickly detect an earthquake. The traditional seismic event identification algorithms mainly include the short-term average/long-term average ratio (STA/LTA) method [27,28,29,30,31], Akaike Information Criteria (AIC) proposed by Akaike [32], and a series of algorithms based on AIC [33,34,35,36,37]. The STA/LTA algorithm utilizes the ratio of the seismic signal to the background noise to judge the occurrence of an earthquake. Since the arrival of the seismic signal will make the ratio change, if it exceeds the preset threshold, the conclusion can be drawn from this that an earthquake has occurred. AIC is a criterion to measure the good fit of statistical models. In seismology, the minimum point of the AIC curve is judged as the arrival time of an earthquake phase [38].

In the past few years, deep learning has been developed swiftly with neural network algorithms as the core. Using the information transfer concept between neurons for summarizing data, we imitate the learning process of the human brain through neural networks, then acquire the complex relationships between input features and output results. Finally, we can make predictions on the new input feature based on the trained models. With these advantages, seismologists have gradually introduced neural networks into their works. For example, Zhang et al. [39] proposed a single-station machine learning algorithm based on Convolutional Neural Networks (CNN) to provide timely and accurate seismic alerts for potential users; Bilal et al. [40] suggested a deep learning model with Graph Neural Network (GNN) and CNN for early earthquake detection; van den Ende and Ampuero [41] realized seismic source characterization through GNN with spatial information; Wang et al. [42] applied a Long Short-term Memory (LSTM) neural network to on-site EEW; Abdalzaher et al. [43] presented a model named three seconds AE and CNN (3S-AE-CNN) to determine magnitude and location of an earthquake; Zhu et al. [44] used a deep CNN to estimate earthquake magnitude; and Perol et al. [45] proposed ConvNetQuake, a kind of CNN for earthquake detection and location from a single waveform with high scalability.

As the studies above mentioned, neural networks can be used effectively in the seismology research field. However, these works are concentrated on using neural network algorithms to process traditional seismic station data, rather than data recorded by mobile phones. Moreover, these network models are not intended for application in EEW and cannot meet its requirement of high effectiveness. Currently, there are few studies on using neural network algorithms for EEW on mobile phones, such as MyShake [23]. In order to make better use of mobile phones to carry out EEW, it is necessary to conduct accurate and efficient earthquake event detection on smartphones. Since the smartphone is different from the seismic station and its location is not fixed, the collected data are greatly affected by non-seismic events, such as human activities and traffic operations. Therefore, traditional seismic event detection algorithms (STA/LTA and AIC) are not fully applicable to mobile phone seismic event identification. Additionally, the accuracy of current algorithms of earthquake event recognition on mobile phones is not high and can be improved. For example, the earthquake event recognition accuracy rate of MyShake in testing [25] and the method under the non-fixed attitude of the mobile phone proposed by Li [46] are above 90%.

In order to improve the accuracy of earthquake event recognition on mobile phones, we explore the application of neural networks in earthquake event recognition on mobile phones in this study. The rest of this paper is organized as follows. Section 2 details the dataset, including data source and preprocess. Section 3 describes our methods for feature selection, neural network models, and their performance. Section 4 compares the models’ performance in this study, then discusses their weaknesses and the limitations of using mobile phones for EEW. Section 5 concludes the paper.

## 2. Datasets

### 2.1. Data Sources

The data used in this study can be divided into two types: non-seismic data and seismic event data. The non-seismic data came from the three-component acceleration data collected by the self-developed mobile phone APP named EEWSensor with a sampling rate of 100 Hz. The interface of this APP is shown in Figure 1. Considering influence factors such as human activities and positions of mobile phones, we installed this APP on three models of smartphones and collected acceleration data under varied conditions, such as various human activities (e.g., running and walking), different speeds (e.g., fast and low), different mobile phones placements (e.g., in hands or bags), varied data lengths (e.g., 30, 60 s). The detailed classification is listed in Table 1. Different from the non-seismic data, we first pre-collected the typical mobile phone sensor noise data when the phone was stationary for the seismic event data. Because a typical mobile phone mems sensor has a resolution of about 1 cm/s^2^ [47], the difference can be ignorable, so the noise amplitude is almost the same. Then, by fusing the traditional strong earthquake waveforms recorded from the KiK-NET and K-NET stations in Japan from January 2021 to March 2021 and the self-noise of mobile phone sensors, the seismic event data was acquired, which was viewed as the seismic data recorded by the EEWSensor. Steps to acquire the simulated data are presented in the next Section. The magnitude distribution of seismic event data is displayed in Figure 2; only about 1% comprises small earthquake events (magnitude < 3). With this method, a total of 52,590 pieces of data were collected, and about 60% were non-seismic data.

### 2.2. Data Preprocessing

The data preprocess here refers to steps required from processing the original data to obtaining the feature data for inputting into the neural network. Since the original formats of non-seismic events and seismic events are dissimilar, the preprocess is different. The whole process is shown in Figure 3. For non-seismic data collected by mobile phones, the preprocessing steps include the following:Removing irrelevant information such as time and location to leave only three-component acceleration;Truncating each acceleration record into specified lengths according to different activities (Table 1).As to seismic data, the preprocessing steps are:Extracting the three-component acceleration data from original files, and merging them into one;Manually excluding those data with unclear *P* wave arrival;Detrending each acceleration;Adding the mobile phone self-noise to each record for simulating earthquake data recorded by a mobile phone.

As the accelerometer self-noise level of a mobile phone is several orders of magnitude higher than a traditional force-balance sensor, the original record noise is almost completely covered after adding. Then we can consider the seismic data recorded directly by the mobile phone. In addition, since the amplitude-frequency characteristics of the acceleration sensor on the mobile phone are basically the same as those of the traditional accelerometer, the difference in the frequency band can be avoided [9]. With this method, we can speedily obtain a large number of seismic event data equivalent to that recorded by mobile phone sensors. Figure 4 shows the comparison of seismic observation records before and after adding the mobile phone sensor noise.

Since most of the non-seismic data were similar to each other, and part of the seismic event data did not meet our requirements, such as without clear *P*-wave phases, after data preprocessing, we finally picked a number of 11,122 pieces of available data, including 5561 pieces of seismic data and 5561 pieces of non-seismic data. These data will be calculated to obtain the features as input of neural network models. The ratio of the training set versus the test set is 7:3, which means 7784 pieces of data in the training set and 3338 pieces in the test set. Each dataset has the same number of seismic events and non-seismic data. The training set will be randomly divided into ten parts, and trained models will be then validated by 10-fold cross-validation with one of the parts.

## 3. Methods

Before formal training, we determined how many and which features could lead to the best performance, detailed in Section 3.1. Then we constructed two kinds of neural network models, Fully Connected Neural Network (FCNN) and CNN. Considering that the amount of data was not large, the 10-fold cross-validation was applied to dividing the training set to reduce the possibility of model overfitting and obtain a higher accuracy of identifying non-seismic data and seismic events.

### 3.1. Feature Selection

The user experience needs to be considered for effective EEW on the mobile phone. Therefore, memory usage and calculation efficiency should be taken into account. For these reasons, there should not exist too many feature types. In the light of the acceleration data, we attempted to calculate 18 features, including the peak acceleration of a frame, the average acceleration in a frame, the absolute median deviation (MAD), interquartile range (IQR), variance, standard deviation, average time-domain energy of the three components, and average time-domain energy of the two horizontal components. Their calculation formulas are detailed in Table 2.

Through calculating correlation coefficients (CCs) between 18 features, we found that there existed relatively high CCs between some features, especially between any two of the IQR, variance, and standard deviation of three components, and the highest could reach 0.96. To avoid strong correlations between the features, we used two approaches to reduce the feature dimensionality for lowering training time and improving the generalization ability of trained models. One method is to manually select the best-fitted features according to training results through permutation and combination; the other is to reduce dimensionality based on the Scikit-learn (SKlearn) Principal Component Analysis (PCA) algorithm [48]. With the PCA algorithm, the feature dimension can be set through parameters. Its Python code is displayed in Algorithm 1.
**Algorithm 1** PCA code in PythonFrom sklearn.decomposition import PCAestimator = PCA(*n_components* = 10) # Initialization, *n_components* is the dimension reduction dimension# Use the training features to determine the orientation of the 10 orthogonal dimensions and transform the original training featurespca_X_train = estimator.fit_transform(data_train_x) # The test features are also transformed according to the above-mentioned 10 orthogonal dimension directions (transform)pca_X_test = estimator.transform(data_test_x)

The main idea of the PCA algorithm is to convert multiple original features into a few principal components (PCs) that can reflect the main characteristic of the data, but instead of simply extracting a few from multiple features. A linear combination of these features is taken to acquire uncorrelated PCs. Each PC corresponds to a variance, and the PCs are sorted in descending order according to the corresponding variance value. The variance of the first PC is the largest, and the variance of the last PC is the smallest [48].

The two methods have their own strengths and weaknesses. Although the first method is time-consuming in that it needs to select features according to the training results of the neural network and correlation relationships, we can directly determine which features are selected. On the contrary, the PCA algorithm takes less time and we can easily modify the parameter settings. However, it is impossible to clearly find which features are picked.

With the manual selection, we trained FCNN models for determining which features would lead to better results. Finally, we picked the five best features. They were the peak acceleration in a frame, the MAD and the variance of the UD component, the average time-domain energy of the UD component, and the average time-domain energy of two horizontal components. Figure 5 displays the results of the five feature CCs. One can see from the figure that only the CC value of the UD MAD and the UD average time-domain energy was relatively higher, approximately 0.82. After excluding any one of them, the performance of the trained model was not as good as the one that retained both, so these two features were kept.

For the PCA algorithm, we tried to reduce high-dimension features to different low dimensions, including 10-, 8-, 5-, and 3-dimensional in FCNN models. The training results of different dimensions are shown in Table 3. Except for the 10-dimensional, the best test accuracies of models in other dimensions were over 99%. The overall performance was the best when reduced to 8-dimensional.

Comparing the training results of the two models after dimensionality reduction, the model using the PCA algorithm to obtain features for training and validation datasets had a worse performance than the first method, as is the generalization ability. Thus, manual selection to determine features was applied to select features. We selected the five features as the input of models.

### 3.2. FCNN

#### 3.2.1. Architecture

FCNN is the most basic type of neural network [49], including an input layer, several hidden layers, and an output layer. To determine a model with the most outstanding performance, we designed several FCNN models with different structures, including three, four, five, and six hidden layers. One example is the structure of the model with three hidden layers, shown in Figure 6. We input a tensor of size 1 × 5 each time, corresponding to five features; the number of nodes in the three hidden layers activated by the Relu function is six, four, and three, respectively; the last one is the output layer with Softmax activation function for calculating the classification probability, and the result is a tensor of size 1 × 2 in the form of (a b). If a > b, it means that the one-hot encoding is (1 0), and the classification result is 0, representing non-seismic data. On the other hand, it is an earthquake event.

#### 3.2.2. Performance

The batch size of FCNN models was set to 64. A dynamic learning rate was set so that the learning rate would decay exponentially with the increase of training steps for models’ higher stability. In addition, we used the binary cross entropy loss function and the Adam optimizer.

For per FCNN model defined, we trained 100 epochs, and the model with the best comprehensive performance was picked as the ultimate one. The accuracy rates of several different networks were all above 0.97. When the hidden layers were four or more, the average validation and the best test accuracy rates reached 0.9. However, the accuracy did not improve significantly with the increased training time, as depicted in Table 4. Therefore, the FCNN model with four hidden layers was eventually adopted. An example of the resulting curves is shown in Figure 7, indicating that the FCNN has an excellent performance in earthquake event recognition on mobile phones.

### 3.3. CNN

#### 3.3.1. Architecture

CNN has superb performance in visual processing [50]. A typical CNN includes three main types of layers: convolutional, pooling, and fully connected layers. The convolutional layer is used to extract data features, the pooling layer is for reducing the dimension of the data to prevent overfitting, and the fully connected layer is used for flattening the output of the final pooling layer [51]. In this work, we only selected five features, so no pooling layer was added to the CNN [52]. Figure 8 presents one of the tested CNN structures, which has two convolution layers and one fully connected layer. Generally, images are taken as the input of CNN. However, here we directly input the five features calculated from three-axis acceleration data into the models. Each feature data of size 1 × 5 in the text files is transferred into a matrix of size 5 × 5. The convolution layer 1 has six convolution kernels of size 3 × 3 with one channel, the layer 2 has ten kernels of size 3 × 3 with six channels. Then, the results after convolution were input into the fully connected layer with shape (10, 6) using the Softmax activation function to calculate the classification result.

#### 3.3.2. Performance

When the batch size of the convolutional neural network was 64, the loss curve of the validation set tended to be unstable, and it was prone to large oscillations, as shown in Figure 9. Therefore, we tried to increase it and ultimately set it to 128. The rest of the parameter settings were consistent with FCNN. The average accuracy of the model on the validation set was 99.32%, and the best test accuracy was 99.34%. Its training and validation curves are shown in Figure 10, demonstrating that CNN can also be well applied to detect seismic events.

### 3.4. Other Algorithms

In addition, we tried two additional machine learning algorithms: logistic regression and random forest. The same five features were used as input for these two models. The average validation accuracy of logistic regression was 95.2%, and its best test accuracy was 96.2%; the random forest was 99.5% and 99.4%, respectively. In evaluating the generalization ability of models, the two algorithms were better at non-seismic data detection, and the accuracy of the logistic regression model on seismic events was higher than that of the random forest. However, both prediction accuracies were less than 90%, significantly lower than FCNN and CNN.

## 4. Discussion

### 4.1. Comparison of the Results of Different Algorithms

The algorithms mentioned above could effectively identify earthquake events, and the training accuracies on the dataset could reach 95%. From the training and verification results, the accuracy of the random forest was not much different from that of the two neural networks; both were more than 98%, while the logistic regression accuracy was less than 96%. As to the generalization results, the overall performance of FCNN was the best, followed by CNN. Nevertheless, the results of the logistic regression and random forest were unstable, in which most seismic events would be identified as non-seismic data. Consequently, it would be prone to false negatives in practical application. From the resulting curves presented in Figure 7 and Figure 10, the training and verification accuracies of two neural networks gradually increased and converged with epochs, having almost the same trends. In addition, the losses were gradually decreasing and converging with epochs, but the trend of resulting curves of CNN on the validation set had relatively large oscillation, not as stable as FCNN (Figure 9). After increasing the batch size, the trend had been improved with slight variation (Figure 10). Compared with the random forest used by Li [46], we selected fewer features, and the average accuracy of neural networks with 10-fold cross-validation was much higher and exceeded 99%, which was also higher than the MyShake result in testing [25].

As a result, the constructed models could identify most seismic events, while only a small part was recognized as non-seismic data, which meant that this part of the seismic data was still indistinguishable, and their features had not been learned. In terms of overall performance, neural networks were more well-matched for the selected dataset.

### 4.2. Can We Apply the Trained Models to an Actual EEW APP?

Although the performance of these models is excellent, they can be further improved from the perspectives of data and efficiency. In terms of a data perspective, the non-earthquake data collected by mobile phone sensors only include six typical kinds of human activities, which cannot fully represent all human behaviors in real life. It is also a difficult task to establish a dataset with enough samples that can cover various situations. Seismic events are obtained from the traditional Japanese strong-motion observation records, not by mobile phone sensors, which may also have an impact on results. In the future, we will attempt to collect more seismic event data by shaking tables [23] or real mobile phones. In addition, we also consider the method proposed by Zheng et al. [53], in which the station data can be directly converted into mobile phone quality data.

From the point of efficiency, through completing the entire process of loading the model and making predictions on new data, the time-consuming on a computer (Inter Xeon E5-2609 v4 CPU Processor 8 Core) was obtained to assess the time-consuming on a smartphone. The average processing time of the computer for each 1 s acceleration data was 4 ms. From this, the average processing time per second of the mobile phone would be less than 50 ms. In addition, a multi-layer algorithm model can be adopted, combining traditional seismic event recognition methods with neural networks or other machine learning algorithms [23,46]. For example, the STA/LTA method is used at the low level of the algorithm to exclude most of the data unrelated to an earthquake, and apply the neural network model to the high level, for dividing the rest data more finely. With this method, the complex calculation of a large amount of data is avoided, further reducing the load of the mobile phone.

The neural network model will be applied to EEWSensor as soon as possible, and be improved and the algorithm optimized according to the actual performance.

### 4.3. Limitations and Corresponding Solutions of EEW on Mobile Phones

Although neural networks hold great promise for seismic event recognition on mobile phones, there exist the following limitations:The sensors in mobile phones are not dedicated to detecting earthquakes, so they cannot reach the accuracy of professional strong-motion accelerometers;The installed capacity of mobile APPs is determined by the results of user experience, such as power consumption, CPU load, and local storage space occupied. If the installed capacity is insufficient, we will not be able to achieve the goal of trading quantity for quality, making it difficult to carry out EEW.

For the former limitation, we can trade quantity for quality since many smartphones exist. With respect to an earthquake, we can collect enough seismic data by mobile phones in a fixed-size region, relative to the traditional accelerometers. Then these data can be overlapped for improving the quality through the array processing method proposed by Inbal et al. [54,55,56]. As to the latter, we can further increase functionality of the APP, not only using it as a data source for acceleration recording, but also receiving seismic information transmitted from the server. Additionally, we should also reduce the complexity of the algorithms, selecting some features that may be more suitable, like the Fourier spectrum mentioned by da Silva et al. [57], making the computing efficiency of the mobile phone better, and improving the user experience with much lower battery consumption. It is also possible to try to embed the EEW function into the operating system of a smartphone [25], or into the APPs with a mass of users, such as WeChat (https://www.wechat.com/, accessed on 13 October 2022) and Facebook (https://www.facebook.com/, accessed on 13 October 2022). In this way, we do not need to consider the installed capacity, and more users will use the function without downloading another APP.

## 5. Conclusions

In this work, mobile phones were employed as data collectors for real-time earthquake recording through using built-in MEMS sensors. We investigated several artificial intelligence algorithms for earthquake detection on the collected data. In terms of feature selection, we compared two means, PCA and manual selection, each with its excellence and shortcomings. The manual selection was ultimately used to determine the best features.

For the reduction of the model overfitting, we used 10-fold cross-validation to train, validate, and test FCNN and CNN models. The results demonstrated that the average recall rate of non-seismic data and the average precision of seismic events both could reach 99%, indicating that these two neural network models are able to distinguish the majority of human activities and earthquake events. With these models, we can effectively improve the accuracy of earthquake event recognition on mobile phones. In the near future, we will focus on collecting more types of data and selecting more appropriate features to train models with better generalization ability.

## Figures and Tables

**Figure 1 sensors-22-08769-f001:**
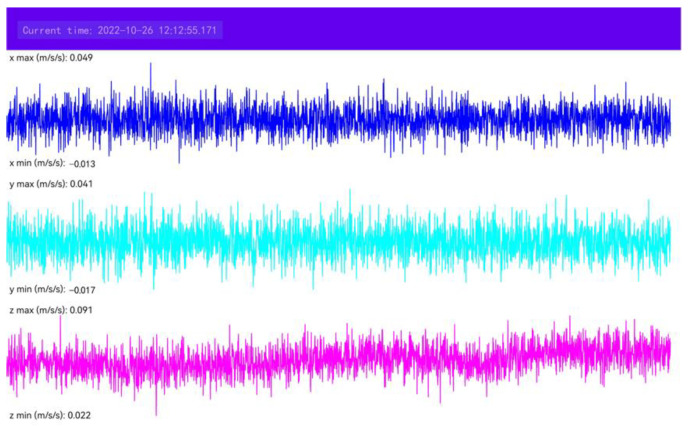
Interface diagram of EEWSensor.

**Figure 2 sensors-22-08769-f002:**
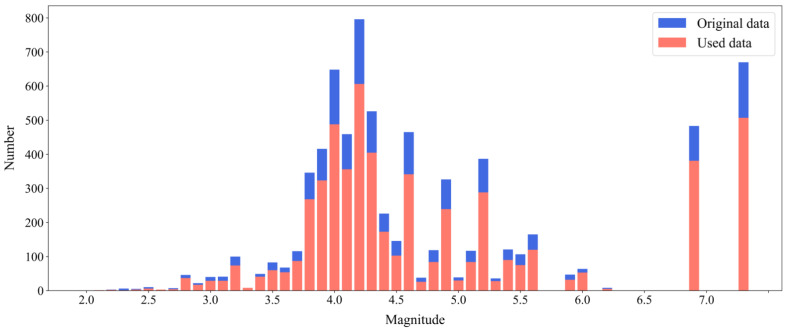
Histogram of the number of selected traditional strong-motion records as a function of magnitude in this paper.

**Figure 3 sensors-22-08769-f003:**
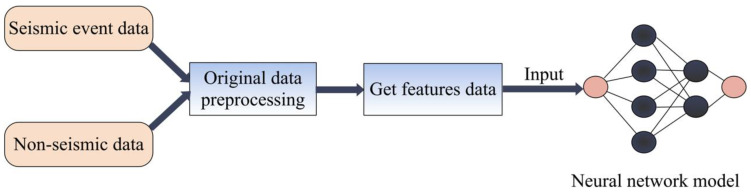
The diagram of data preprocessing.

**Figure 4 sensors-22-08769-f004:**
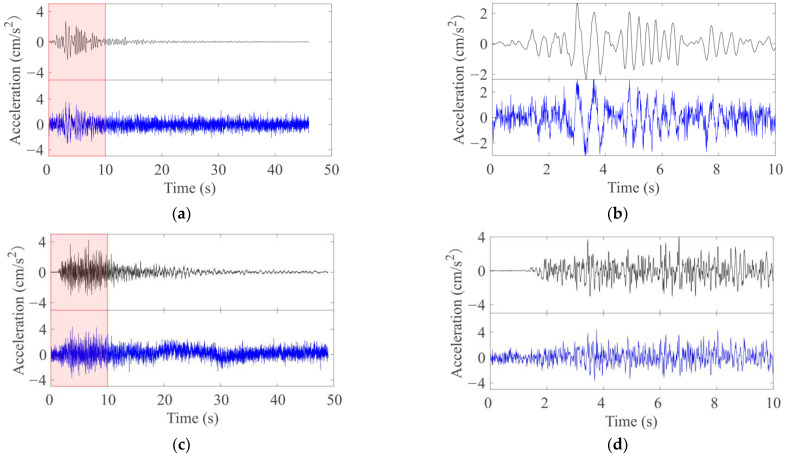
Comparison of seismic event acceleration waveforms before and after adding mobile phone sensor noise. The left column is the UD component of each earthquake event acceleration waveform, and the right column is the enlarged part of the corresponding left shadow area (0–10 s). The black line of each row is the data before adding noise, and the next blue line is after adding noise to the previous. (**a**,**b**): Near Earthquake, *M*4.7, Δ=103.1 km, Station: SZO018; (**c**,**d**): Local Earthquake, *M*4.7, Δ=26.3 km, Station: TYK009; (**e**,**f**): Near Earthquake, *M*7.3, Δ=830.9 km, Station: SMNH16; (**g**,**h**): Local Earthquake, *M*7.3, Δ=68.1 km, Station: FKSH20. *M* is the magnitude and Δ is the epicentral distance.

**Figure 5 sensors-22-08769-f005:**
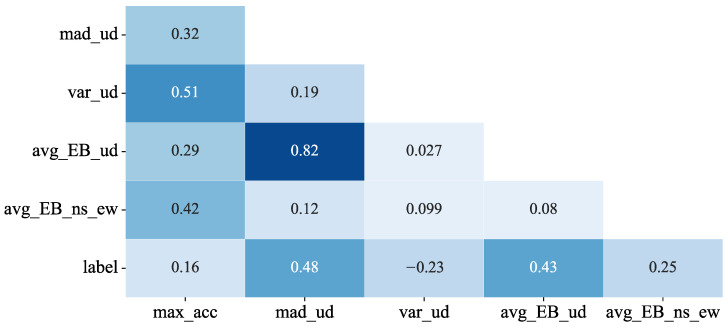
Heatmap of the feature correlation coefficients. max_acc: the peak acceleration in a frame; mad_ud: the MAD of the UD component; var_ud: the variance of the UD component; avg_EB_ud: the average time-domain energy of the UD component; avg_EB_ns_ew: the average time-domain energy of two horizontal components; label: the data label. If it is 1, the data is an earthquake event, and non-seismic data if it is 0.

**Figure 6 sensors-22-08769-f006:**
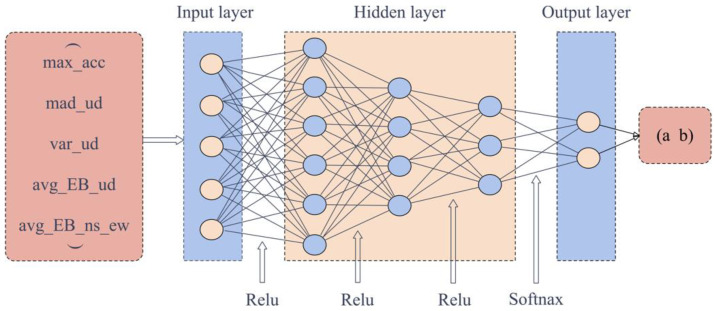
FCNN architecture with three hidden layers, including three hidden layers and an output layer. Max_acc, mad_ud, var_ud, avg_EB_ud and avg_EB_ns_ud are the five features, consistent with the description in Figure 5. (a b) is the classification result.

**Figure 7 sensors-22-08769-f007:**
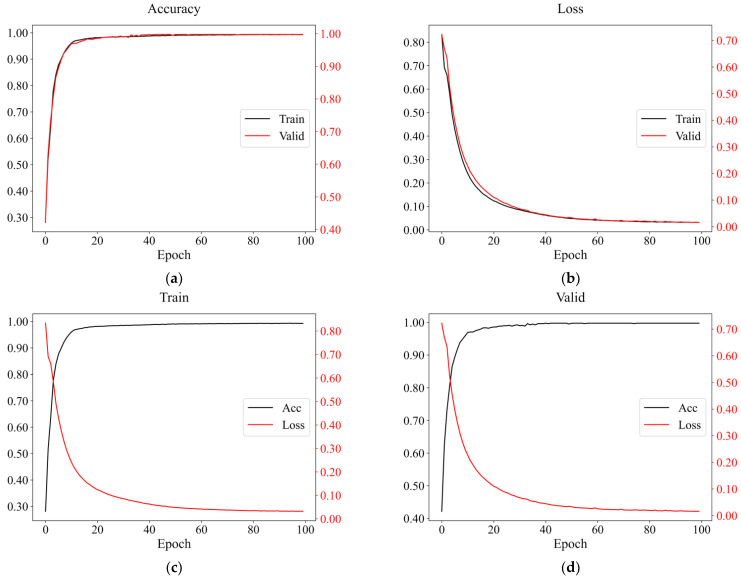
An example of the resulting curves of a four-layer FCNN when *K* = 7. (**a**) The accuracy and (**b**) the loss curves of the training and validation datasets; the accuracy and loss curves of (**c**) the training and (**d**) the validation datasets. *K* represents which fold.

**Figure 8 sensors-22-08769-f008:**
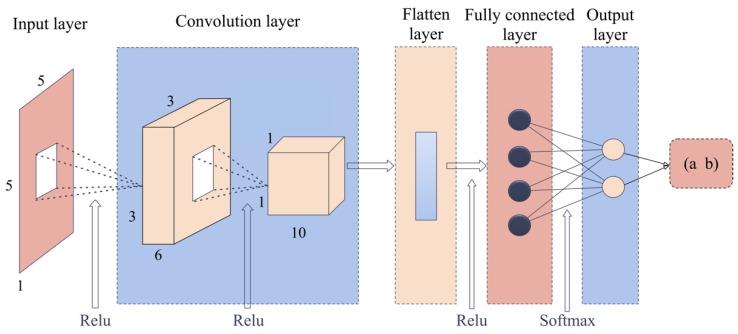
One of the Convolution Neural Network (CNN) architectures, including the input layer, two convolution layers, one fully connected layer, and the output layer.

**Figure 9 sensors-22-08769-f009:**
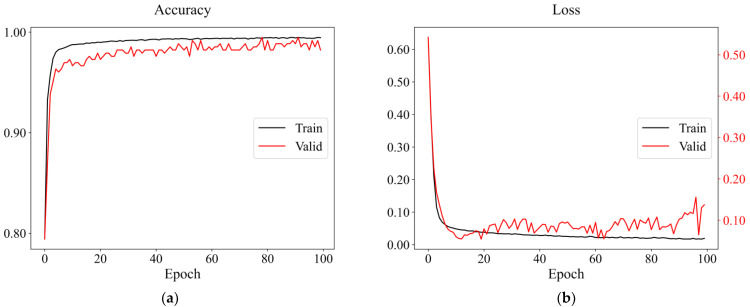
Curves of different folds of CNN on the validation set when the batch size was 64. (**a**) The accuracy curve of *K* = 7; (**b**) the loss curve of *K* = 8.

**Figure 10 sensors-22-08769-f010:**
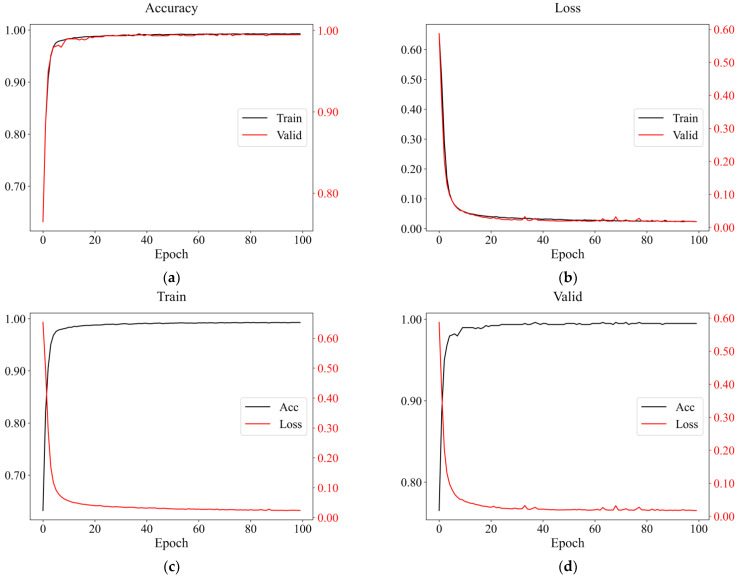
An example of resulting curves of CNN when *K* = 2. (**a**) The accuracy and (**b**) loss curves of the training and validation set; the accuracy and loss curves of (**c**) the training and (**d**) validation set.

**Table 1 sensors-22-08769-t001:** Details of mobile phone model and data type.

Phone Model	Data Type	Phone’s Location	Velocity	Data Length (Second)
Vivo S5 Huawei nova 5pro Xiaomi 8se	Walking	In hands/In bags	Low/Normal/Fast	90
	Running	In hands/In bags	Low/Fast	
	Going up and down stairs	In hands/In bags	Normal	
	Riding a bike	In bags		
	Taking the bus	In hands/In bags		30/35/60/90
	Taking the subway	In hands/In bags		

**Table 2 sensors-22-08769-t002:** Calculation formulas of the selected features in this study. If the number of features is three, it means that this feature will be calculated separately for three components of acceleration data.

Feature Name	Number of Features	Calculation Formula	Explanation
The peak acceleration in a frame	1	$\max(\sqrt{x_{i}^{2}+y_{i}^{2}+z_{i}^{2}}$)	*x*, *y*, *z*: EW, NS, UD at a certain moment *i*: index, the value is [1, *N*] $Q_{3},Q_{1}$: the third and the first quartile *S*: a component data, such as EW, NS, or UD *N*: data length μ: average of UD
The average acceleration in a frame	1	$\frac{1}{N}\overset{N}{\underset{i=1}{\sum }}\sqrt{x_{i}^{2}+y_{i}^{2}+z_{i}^{2}}$	
The absolute median deviation	3	median(|S−median(S)|)	
Interquartile range (IQR)	3	$IQR=Q_{3}-Q_{1}$	
Variance	3	$\frac{1}{N}\overset{N}{\underset{i=1}{\sum }}{\left(S_{i}-\mu \right)}^{2}$	
Standard deviation	3	$\sqrt{\frac{1}{N}\overset{N}{\underset{i=1}{\sum }}{\left(S_{i}-\mu \right)}^{2}}$	
Average time-domain energy	3	$\frac{1}{N}\overset{N}{\underset{i=1}{\sum }}S^{2}$	
Horizontal average time-domain energy	1	$\frac{1}{N}\overset{N}{\underset{i=1}{\sum }}x_{i}^{2}+y_{i}^{2}$	

**Table 3 sensors-22-08769-t003:** Performances of Fully Connected Neural Networks (FCNNs) with different feature dimensions of PCA.

Feature Dimension	Average Validation Accuracy	Best Test Accuracy
10	78.98%	97.33%
8	78.92%	99.25%
5	78.7%	99.04%
3	77.39%	99.22%

**Table 4 sensors-22-08769-t004:** Performances of FCNN models with different hidden layers.

Hidden Layers	Number of Nodes	Average Validation Accuracy	Best Test Accuracy	Training Time (Second)
2	4, 3	98.77%	98.4%	150
3	6, 4, 3	99.06%	98.83%	196
4	8, 6, 4, 3	99.23%	99.46%	246
5	10, 8, 6, 4, 3	99.27%	99.36%	291
6	12, 10, 8, 6, 4, 3	99.29%	99.37%	360

## Data Availability

The data that support the findings of this study are available from the corresponding author, M.C., upon reasonable request.

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
