# Peer review of "Earthquake Event Recognition on Smartphones Based on Neural Network Models"

_sensors, 2022, doi:10.3390/s22228769_

Round 1
Reviewer 1 Report
The authors have developed an Earthquake Event Recognition methodology using Neural Network Models that can be used for detecting earthquakes from Micro-Electro-Mechanical systems in mobile devices. The paper is quite interesting, the results of accuracy and precision are very good, and the questions I have about the paper concern the methodology of data treatment and statistics.
The authors have spent a lot of effort manipulating distinct “noise” types to add to the earthquake signal, for instance: walking, running, going up and down stairs, cycling, being inside the bus or the railway. However, if the Earthquake Event Recognition methodology using mobile devices would been used in the future, probably it will be based in thousands or even hundred of thousands of mobile signals, depending on the city. In this case, considering a probability of 20% of the mobiles being at rest on the table this would be already a large amount of high quality data to be used as input in the IA Earthquake Event Recognition machine.
The features explored in the paper are computed over a time window. If a time window is used in the methodology why it was not used any Fourier spectral measure as a feature? I suggest the authors to use spectral measure in future works, see for instance the paper: da Silva, S. L. E. F.; Corso, G., Microseismic event detection in noisy environments with instantaneous spectral Shannon entropy. PHYSICAL REVIEW E., v.106, p.014133 - , 2022. By the way, this work could be cited in the manuscript, it is brand new methodological paper to investigate earthquake event detection.
In the line 189 are cited 18 features employed in the methodology, in table 2 only seven features are indicated, what are the missing ones? Figure 5 cited the five features most uncorrelated ones. I understood that this 5 features are used as the input to the Neural Network Model to perform the earthquake event recognition. Please tell me if I am wrong, this is not completely clear in the text. Most of the earthquake event prediction methodologies used the full time series to perform this task, not a couple of time series features. The features are additional variables of the neural network model or the input of the neural network model?
The magnitude of the seismic event acceleration data in figure 4 ranges from 0.1 gal to 300 gal. The added noise to the data have always the same amplitude? I expect that it is much easier to detect a true seismic signal in the situation where the seismic signal have a large amplitude (Fig 4g) than in the case of small amplitude (Fig. 4f). What is the impact of this point in the accuracy of the results? That means, if one drops out the small earthquakes the accuracy of the network model will significantly improve?
Minor points:
1) In figure 6 after the input box, under the arrow, it is written ?x5. Please make clear the meaning of the “?” signal. The five corresponds to the five used features, but the interrogation is unclear.
2) The same problem is present in figure 7. Please make clear the “?x5x5x1”
3) The output of figure 6 is “dense_305”, is this an abbreviation or a code? What is the meaning of 305. The same question is extended to the output of figure 7, what means “dense_379”.
4) The acceleration unit measure 1 gal which corresponds to 1 cm/s2 , this unity is not common in physics. I suggest to insert in the legend of the figure, or in the text, this definition.
Author Response
Response to Reviewer 1 Comments
The authors have developed an Earthquake Event Recognition methodology using Neural Network Models that can be used for detecting earthquakes from Micro-Electro-Mechanical systems in mobile devices. The paper is quite interesting, the results of accuracy and precision are very good, and the questions I have about the paper concern the methodology of data treatment and statistics.
Point 1: The authors have spent a lot of effort manipulating distinct “noise” types to add to the earthquake signal, for instance: walking, running, going up and down stairs, cycling, being inside the bus or the railway. However, if the Earthquake Event Recognition methodology using mobile devices would be used in the future, probably it will be based on thousands or even hundreds of thousands of mobile signals, depending on the city. In this case, considering a probability of 20% of the mobiles being at rest on the table this would be already a large amount of high-quality data to be used as input in the IA Earthquake Event Recognition machine.
Response 1:
We are sorry for this misunderstanding. The added noise is the data collected when the phone is at rest on the table without any disturbance. The corresponding description is added now: “we first pre-collected the typical mobile phone sensor noise data when the phone was stationary for the seismic event data.”
The data, such as walking, running, going up and down stairs, cycling, and being inside the bus or the railway, are non-seismic data, not the noise added to the earthquake signal. These data will be firstly processed on mobile phones with our proposed methods. And then when the data are detected as an earthquake, they will be transferred to the server. Therefore, only three-component acceleration data is processed on a smartphone at the same time.
Point 2: The features explored in the paper are computed over a time window. If a time window is used in the methodology why it was not used any Fourier spectral measure as a feature? I suggest the authors to use spectral measure in future works, see for instance the paper: da Silva, S. L. E. F.; Corso, G., Microseismic event detection in noisy environments with instantaneous spectral Shannon entropy. PHYSICAL REVIEW E., v.106, p.014133 - , 2022. By the way, this work could be cited in the manuscript, it is brand new methodological paper to investigate earthquake event detection.
Response 2:
The trained models will be used on the mobile phone. For lowering the CPU load of a mobile phone, we selected the 18 features described in the paper.
As suggested, we will try to use the Fourier spectral measure as a feature in the future and have added it to the discussion accordingly.
Point 3: In the line 189 are cited 18 features employed in the methodology, in table 2 only seven features are indicated, what are the missing ones? Figure 5 cited the five features most uncorrelated ones. I understood that this 5 features are used as the input to the Neural Network Model to perform the earthquake event recognition. Please tell me if I am wrong, this is not completely clear in the text. Most of the earthquake event prediction methodologies used the full time series to perform this task, not a couple of time series features. The features are additional variables of the neural network model or the input of the neural network model?
Response 3:
Since the acceleration data has three components (EW, NS and UD), some features are calculated three times using the same formula with different component data. These features are the absolute median deviation, interquartile range, variance, standard deviation and average time-domain energy. For clarification, a column is added in Table 2 for describing the number of features.
The model designed is for earthquake event detection on a mobile phone and for EEW purposes, so the full-time series features are not suitable. Because they may significantly increase the CPU load of a mobile phone and lower the performance of real-time processing.
The features are the input of the neural network model. A sentence is added: “ And we selected the five features as the input of models.”
Point 4: The magnitude of the seismic event acceleration data in figure 4 ranges from 0.1 gal to 300 gal. The added noise to the data have always the same amplitude? I expect that it is much easier to detect a true seismic signal in the situation where the seismic signal have a large amplitude (Fig 4g) than in the case of small amplitude (Fig. 4f). What is the impact of this point in the accuracy of the results? That means, if one drops out the small earthquakes the accuracy of the network model will significantly improve?
Response 4:
Yes, the amplitude of the added noise is almost the same since the noise data is collected when the mobile phone is stationary. The MEMS sensor resolutions of typical smartphones are approximately 1 cm/s2, and the difference can be ignorable. The corresponding description is added: “Different from the non-seismic data, we first pre-collected the typical mobile phone sensor noise data when the phone was stationary for the seismic event data. Because a typical mobile phone mems sensor has a resolution of about 1cm/s2 [47], the difference can be ignorable, so the noise amplitude is almost the same. Then, by fusing the traditional strong earthquake waveforms recorded from the KiK-NET and K-NET stations in Japan from January 2021 to March 2021 and the self-noise of mobile phone sensors, the seismic event data was acquired, which could be viewed as the seismic data recorded by the EEWSensor. ”
We agree with you that “it is much easier to detect a true seismic signal in the situation where the seismic signal has a large amplitude (Fig 4g) than in the case of small amplitude (Fig. 4f)”. There only exists a small fraction (1%) of small earthquake event data, as seen in figure 2 (magnitude < 3). So if we exclude this part, the accuracy will not be influenced significantly. In addition, for an EEW purpose on mobile phones, we also concern more about an earthquake with a large magnitude.
The corresponding content has been modified.
Minor points:
- In figure 6 after the input box, under the arrow, it is written ?x5. Please make clear the meaning of the “?” signal. The five corresponds to the five used features, but the interrogation is unclear.
- The same problem is present in figure 7. Please make clear the “?x5x5x1”
Response 5:
“?” indicates how much data there are in figure 6 and 7. Each data contains 5 features, so each input data of FCNN models is in the shape of 1x5. For better computation of the CNN, we convert the size of 1x5 to 5x5. Also, since it is non-image data, the number of input channels is 1. Therefore, each input shape of CNN models is 1x5x5x1.
For clarification, we have redrawn Figures 6 and 7.
- The output of figure 6 is “dense_305”, is this an abbreviation or a code? What is the meaning of 305. The same question is extended to the output of figure 7, what means “dense_379”.
Response 6:
Generally, “dense_XX” means that this is the XXth layer. While in this work, due to repeated training multiple times, the number in the default name of the model layer increased, did not mean that there were so many layers in the model. The same as “dense_379”.
For clarification, Figures 6 and 7 are redrawn.
- The acceleration unit measure 1 gal which corresponds to 1 cm/s2 , this unity is not common in physics. I suggest to insert in the legend of the figure, or in the text, this definition.
Response 7:
As suggested, all “gal” in figures have been revised to “cm/s2”.
At last, thank you again for your arduous work and constructive advice. We wish good health to you, your family, and your community. Your careful review has helped to make our study clearer and more comprehensive.

Reviewer 2 Report
In introduction, i would suggest authors to explicitly enhance the rationale of study with extensive review of the past studies and please highlight the gap as well. At the end paragraphs, please mention general and specific objectives of the study and remove all information with Chapters' details--uncommon practice.
In discussion, authors should justify the study results with science and logic. Please make discussion section more justified. Discussion should be more specific to results of the study. Please do as suggested. Else, the manuscript cant be considered for publication. Please enhance all the figures quality as well. Now, i have recommend this manuscript as, "Reject and Resubmit".
Author Response
Response to Reviewer 2 Comments
Point 1: In introduction, I would suggest authors to explicitly enhance the rationale of study with extensive review of the past studies and please highlight the gap as well. At the end paragraphs, please mention general and specific objectives of the study and remove all information with Chapters' details--uncommon practice.
Response 1:
As suggested, the relevant content is revised. The difference between previous studies and what we have done in this work is added in the introduction:
“As the studies above mentioned, neural networks can be used effectively in the seismology research field. However, these works are concentrated on using neural network algorithms to process traditional seismic station data, rather than data recorded by mobile phones. Moreover, these network models are not intended for application in EEW and cannot meet its requirement of high effectiveness. Currently, there are few studies on using neural network algorithms for EEW on mobile phones, such as MyShake [23]. In order to make better use of mobile phones to carry out EEW, it is necessary to conduct accurate and efficient earthquake event detection on smartphones. Since the smartphone is different from the seismic station and its location is not fixed, the collected data is greatly affected by non-seismic events, such as human activities and traffic operations. Therefore, traditional seismic event detection algorithms (STA/LTA and AIC) are not fully applicable to mobile phone seismic event identification. Additionally, the accuracy of exsiting algorithms of earthquake event recognition on mobile phones is not high and can be improved. For example, the earthquake event recognition accuracy rate of MyShake in the testing [25] and the method under the non-fixed attitude of the mobile phone proposed by Li [46] are above 90%.
In order to improve the accuracy of earthquake event recognition on mobile phones, we will explore the application of neural networks in earthquake event recognition on mobile phones in this study…”
As suggested, all information with Chapters’ details is removed in the Conclusion section.
Point 2: In discussion, authors should justify the study results with science and logic. Please make discussion section more justified. Discussion should be more specific to results of the study. Please do as suggested. Else, the manuscript cant be considered for publication. Please enhance all the figures quality as well. Now, i have recommend this manuscript as, "Reject and Resubmit".
Response 2:
As suggested, the discussion part is revised and more specific to the results of the study now. Some information is removed: “only two APPs are used for EEW purpose with the built-in accelerometer. One is MyShake [23-25], and the other is Earthquake Network [18, 26]”.
In addition, we divided the discussion into three subsections to make it more logical. The first subsection compares the results of our work with other algorithms. Then the second discusses whether the existing well-performing models can be applied to the EEWSensor APP from two aspects of data and efficiency. The third subsection explores some limitations and improved methods for EEW using mobile phones.
At last, thank you again for your arduous work and constructive advice. We wish good health to you, your family, and your community. Your careful review has helped to make our study clearer and more comprehensive.
